# Chitosan Oligosaccharides Coupling Inhibits Bacterial Biofilm-Related Antibiotic Resistance against Florfenicol

**DOI:** 10.3390/molecules25246043

**Published:** 2020-12-21

**Authors:** Xianghua Yuan, Jing Liu, Ruilian Li, Junlin Zhou, Jinhua Wei, Siming Jiao, Zhuo A. Wang, Yuguang Du

**Affiliations:** 1College of Life Science, Sichuan Normal University, Chengdu 610101, China; yuanxianghua@sicnu.edu.cn (X.Y.); zjunlin1@163.com (J.Z.); 2State Key Laboratory of Biochemical Engineering, Institute of Process Engineering, Chinese Academy of Sciences, Beijing 100190, China; jingliu0223@126.com (J.L.); rlli@ipe.ac.cn (R.L.); jhwei@ipe.ac.cn (J.W.); smjiao@ipe.ac.cn (S.J.); 3College of Chemistry and Chemical Engineering, University of Chinese Academy of Sciences, Beijing 100049, China

**Keywords:** biofilm, florfenicol, chitosan oligosaccharide, antibiotic resistance

## Abstract

The formation of bacterial biofilms has increased the resistance of bacteria to various environmental factors and is tightly associated with many persistent and chronic bacterial infections. Herein we design a strategy conjugating florfenicol, an antibiotic commonly used in the treatment of *streptococcus*, with the antimicrobial biomaterial, chitosan oligosaccharides. The results demonstrated that the florfenicol-COS conjugate (F-COS) efficiently eradicated the mature *Streptococcus hyovaginalis* biofilm, apparently inhibiting drug resistance to florfenicol. A quantity of 250 μg/mL F-COS showed effective inhibitory activity against planktonic cells and biofilm of the bacteria, and a 4-fold improvement of the F-COS compared to unmodified florfenicol was observed. Furthermore, the conjugate showed a broad-spectrum activity against both Gram-positive and Gram-negative bacteria. It suggested that F-COS might have a potential for application in the treatment of biofilm-related infections.

## 1. Introduction

Swine streptococcosis is a widespread infectious disease of zoonosis. Its high fatality rate leads to substantial economic losses in pig farms [1,2]. At present, the main problem of the treatment of *swine streptococcus* is resistance to antibiotics. Florfenicol is a commonly used medicine to treat swine streptococcosis [3]. It is a new kind of broad-spectrum antibacterial agent successfully developed in the late 1980s [3,4]. Since discovered, the application of florfenicol in the veterinary field has achieved remarkable success. However, drug resistance to florfenicol continues to spread due to the overuse of antibiotics in recent years [5,6,7,8]. Novel strategies such as combination treatment with other drugs or chemical modification to solve antibiotic resistance are urgent to develop.

Moreover, the formation of biofilms aggravates the antibacterial resistance [9,10]. Biofilms are a group of bacteria wrapped with extracellular polysaccharide, lipoprotein, and fibrinogen. Biofilm can be irreversibly attached to surfaces such as surface lesions and medical instruments [11,12]. The formation of biofilms was tightly associated with persistent and chronic bacterial infections, increasing bacterial resistance to various factors [13,14,15,16,17]. Studies have shown that bacteria in biofilms are 10–1000 times more opposed to antibiotics compared to planktonic bacteria [18], which may be due to the decreased penetration of antibiotics [19]. Besides, the low-level metabolism of bacteria in biofilm further improved the resistance to antibiotics [19,20].

Previously, we conjugated an aminoglycoside antibiotics streptomycin with chitosan oligosaccharides (COS) to enhance its antimicrobial activity [21]. COS, the degradation product of chitosan, usually contains 2 to 20 units of β-(1–4)-linked D-glucosamine and *N*-acetyl-D-glucosamine, possessing versatile biological activities such as antimicrobial [22]. The conjugate showed potent activity against *P. aeruginosa* biofilm. This finding suggested that the conjugation of antibiotics with antimicrobial oligosaccharide COS was a potential strategy to improve its activity. However, we only observed the improvement of the antibacterial activity on the mature biofilm but not planktons. Moreover, the conjugate was ineffective in removing the biofilm of the gram-positive organism *S. aureus* [21].

In this study, we conjugated florfenicol with COS to study its influence on antibiotic resistance. Mass spectrometry and NMR analyses proved the successful conjugation between florfenicol and COS. Interestingly, the conjugate effectively killed both planktons and biofilm cells of *Streptococcus hyovaginalis*. The results suggested that modifying antibiotics with COS was a potential strategy to overcome antibiotic resistance, although the influence of COS on antibiotics might be different due to the selection of antibiotics.

## 2. Results

### 2.1. Synthesis of Florfenicol-COS Conjugates (F-COS)

To obtain the conjugation between florfenicol and COS, we first introduced a pendant carboxylic acid via standard coupling chemistry with butanedioic anhydride to produce Flo-COOH (FCOOH). Then the carboxyl group of FCOOH was activated and added to the dissolved COS. The synthetic route is shown in Figure 1.

### 2.2. Characterization of Florfenicol-COS Conjugates (F-COS)

#### 2.2.1. Flo-COOH

The ^1^H NMR spectra of Flo-COOH was shown in Figure 2, ^1^H NMR (500 MHz, DMSO) δ 8.88 (d, *J* = 8.9 Hz, 1H), 7.89 (d, *J* = 8.3 Hz, 2H), 7.61 (d, *J* = 8.3 Hz, 2H), 6.44 (s, 1H), 6.01 (d, *J* = 4.3 Hz, 1H), 4.67-4.21 (m, 3H), 3.19 (s, 3H), 2.77-2.59 (m, 2H), 2.54 (t, *J* = 6.5 Hz, 2H). The peaks from 2.46 to 2.56 indicated the methylene groups of butanedioic acid, suggesting the successful conjugation of florfenicol and butanedioic acid (Figure 2).

#### 2.2.2. F-COS

The conjunction between florfenicol and COS was proved by Matrix-Assisted Laser Desorption/Ionization Time of Flight (MALDI-TOF) Mass Spectrometry analysis. Interestingly, the result showed that each COS oligomer (COS 2-COS 8) conjugated to two florfenicol molecules (Figure 3). Unmodified COS oligomers showed peaks from *m/z* 341.35 to 1326.33 with a 161.0 Da mass difference (GlcN residue) between each two peaks (Appendix A). The spectrum of the F-COS conjugate showed peaks at *m/z* 1218.01, 1379.96, 1541.22, 1701.59, 1862.01, 2184.57 represented COS disaccharides, trisaccharides, tetrasaccharides, pentasaccharides, hexasaccharide, octasaccharides conjugated with two molecules of florfenicol, respectively. The heptasaccharide-florfenicol conjugate was not detected, possibly due to the relatively low content of heptasaccharide in the COS product used in the experiment.

### 2.3. Antibacterial Activity of F-COS Conjugates against S. hyovaginalis Biofilm and Planktonic Cells

We assessed the F-COS conjugate’s antibacterial activity against established *S. hyovaginalis* biofilm using a 96-well microtiter plate assay. For the biofilm inhibition test, MRS medium containing antimicrobial compounds were added at the beginning of the biofilm formation. For the biofilm eradication assay, the compounds were added after 24-h formation of biofilms. The result showed that the antimicrobial activity of F-COS on cells in the biofilm was much higher than that of florfenicol at the same concentration (250 μg/mL) (Figure 4A,B). On the contrary, a combination treatment of florfenicol and COS did not show an apparent biofilm-disruption effect (Figure 4A). Also, the inhibitory activity of F-COS on biofilm formation was tested. Consistently, F-COS showed a noticeable enhanced antimicrobial activity compared to the unmodified florfenicol (Figure 4C,D). Similar effects were observed in the fluorescence microscopy study (Figure 5). Furthermore, the antimicrobial activity of F-COS on planktonic *Streptococcus hyovaginalis* cells was also determined. F-COS strongly inhibited bacteria growth at a concentration of 250 μg/mL (Figure 6A).

### 2.4. Broad-Spectrum Anti-Biofilm Activity of F-COS Conjugates against Different Bacteria

To investigate the spectrum of anti-biofilm activity of F-COS conjugates, we assessed the activity with F-COS on the mature biofilm of another Gram-positive bacteria, *S. aureus*, and two Gram-negative pathogens *P. aeruginosa* and *S.enterica*. The result showed that F-COS had a broad spectrum of anti-biofilm activity against all tested bacterial strains. Mostly, the *S.enterica* biofilm, which showed strong resistance to florfenicol, was highly sensitive to F-COS (Figure 6B). These results suggested that conjugating florfenicol with COS efficiently inhibited the biofilm-related florfenicol resistance of both Gram-positive and Gram-negative bacteria.

## 3. Discussion

In this work, we synthesized florfenicol-COS conjugates by a two-step method leaving the active group of florfenicol unaffected (Figure 1). The method is relatively simple and feasible for further industry scale-up. Mass spectrometry results suggested that each COS oligomer (COS2-COS 8) conjugated to two florfenicol molecules (Figure 3). This may be due to the mole ratio of florfenicol and COS used in the reaction. Notably, COS disaccharide was also attached with two antibiotic molecules suggesting the space resistance is not an essential factor in the reaction. Our data showed that the F-COS conjugate was much more effective in eradicating the established biofilm of *Streptococcus hyovaginalis* (Figure 4 and Figure 5). Furthermore, F-COS showed a broad spectrum of anti-biofilm activity against several tested pathogenic bacteria (Figure 6), both Gram-positive and Gram-negative ones, especially on strains showing resistance to florfenicol. This suggested that F-COS might have a potential for application in the treatment of biofilm-related infections.

A defect in florfenicol is its low water solubility. Several studies were devoted to solving the water-soluble of florfenicol, for example, preparation of florfenicol phosphate, florfenicol succinic acid salt [23,24,25,26]. Here, we also found that the water solubility of F-COS (16 mg/mL) was ten times more than that of florfenicol (1.4 mg/mL). Thus, the conjugation with COS showed a dual improvement in both antimicrobial activity and water solubility of florfenicol. This may suggest a novel route for modifying chloramphenicol antibiotics.

Microbial biofilms were closely related to persistent chronic infections in clinical practices. Studies showed that antibiotic resistance could be obtained by genomic change or biofilm formation [19]. It has been reported that chitosan coupled with streptomycin could dramatically effectively inhibit Gram-positive organisms, since chitosan was an efficient carrier to deliver streptomycin into cells [27]. Previously, we conjugated COS with streptomycin to improve its anti-biofilm activity [21]. The conjugate was much more effective in removing Gram-negative bacterial biofilm than the unmodified antibiotic. COS, as chitosan, has apparent antimicrobial activity, good biocompatibility, and low toxicity. Considering the excellent water solubility of COS compared to chitosan, this finding suggested that the conjugation of antibiotics with COS was a potential strategy to improve its activity. However, both CS-Strep and COS-Strep conjugate showed a limited spectrum of antimicrobial activity. In this article, we synthesized the conjugate of florfenicol and COS, which dramatically inhibits the biofilm-related resistance to florfenicol. A quantity of 250 μg/mL F-COS showed significant anti-biofilm activity, while and florfenicol alone did not effectively eradicate biofilm even at 1000 μg/mL (Figure 4 and Figure 5). Similar results were found in the inhibitory test against planktonic bacteria cells (Figure 6). Furthermore, F-COS showed a broad spectrum of anti-biofilm activity against Gram-positive and Gram-negative bacteria (Figure 6), unlike the COS-Strep conjugate. Thus, compared to CS-Strep and COS-Strep, F-COS exhibited different antimicrobial actions, although they all are beta-1,4-glucosamine glycan-antibiotic conjugates. The broadspectrum activity of F-COS suggested good potential in the practical application.

The possible mechanism for the observed results remains unclear. One possible explanation is that the modification on florfenicol with COS impaired the drug-resistant machinery. Bacteria gain the resistance to florfenicol partially by the drug efflux pump system, leading to reduced drug concentration in the bacteria cells [28]. Our previous study showed that COS modification on streptomycin suppressed the MexX-MexY drug efflux pump system in *P. aeruginosa* [21]. The F-COS conjugate might have a similar action mode. Also, the antimicrobial activity of COS should be considered. Previous studies indicated that COS had sound bactericidal effects, especially against gram-negative bacteria [29]. A simple mixture of florfenicol and COS did not show an apparent synergistic antimicrobial effect (Figure 4), indicating the significant antimicrobial activity of the F-COS conjugate not only due to the combined usage of two drugs. However, membrane disrupting activity of COS might help the F-COS conjugate get into the bacteria cell, resulting in better antimicrobial effects. Moreover, increased water solubility of F-COS compared to florfenicol might increase the effective drug concentration at the bacterial interface, which might also contribute to the enhanced activity of the conjugate.

In conclusion, our findings demonstrate that florfenicol conjugating with COS showed enhanced antimicrobial activity against both planktonic cells and biofilm cells of *Streptococcus hyovaginalis*. The conjugate showed a broad-spectrum inhibitory activity on biofilm-related resistance to florfenicol. COS has received considerable attention as biomaterials for its multiple functions and good biocompatibility. It provides a new strategy for combating drug resistance.

## 4. Materials and Methods

### 4.1. Reagents and Material

Chitosan oligosaccharides (COS) were prepared as previously described with the deacetylation degree over 95% and average molecular weight below 1 kDa [30]. Florfenicol and other reagents were purchased from Sigma (St. Louis, MO, USA). *Streptococcus hyovaginalis* strain (CGMCC1.10866) and *Staphylococcus aureus* strain (CGMCC1.2910) were purchased from the China General Microbiological Culture Collection Center (CGMCC). *Pseudomonas aeruginosa* (PAO1) strain was granted by Prof. Ma Lvyan. S. hyovaginalis, *S. aureus*, and *P. aeruginosa* were cultured in MRS medium at 30 °C, TSB medium at 37 °C or LB medium at 28 °C, respectively.

### 4.2. Synthesis and Purification of Florfenicol-COS Conjugates (F-COS)

#### 4.2.1. Synthesis of Florfenicol Amber Acid Ester (F-COOH)

F-COOH was synthesized based on previously described protocols with minor modifications [3]. The reaction was performed under an N_2_ atmosphere using standard Schlenk reaction protocols. Florfenicol (5.3 g, 15 mmol) and DMAP (0.3 g, 7.5 mmol) were dissolved in 100 mL anhydrous acetone and butanedioic anhydride (2.25 g, 25 mmol) was added to the reaction mixture. The mixture was heated under reflux and stirred overnight. Then, acetone was removed under reduced pressure, yielding a light-yellow viscous oil. The oil was dissolved in ethyl acetate (200 mL) and washed with 1 M HCl (3 × 50 mL) and saturated NaCl (3 × 50 mL). The organic layer was separated, dried over Na2SO4 and concentrated. The F-COOH residue was purified by recrystallization (dichloromethane:ethanol = 95:5) as a white solid (4.42 g, 64.5%).

#### 4.2.2. Synthesis of Florfenicol-COS Conjugates (F-COS)

NHS (0.88 g, 7.8 mmol) was added to the mixture of F-COOH (3.44 g, 7.8 mmol) and EDCI-HCl (1.35 g, 7 mmol) dissolved in anhydrous dichloromethane (~100 mL). The mixture was stirred for 2 h at room temperature and evaporated under reduced pressure, yielding a light-yellow viscous oil. Then the viscous oil was dissolved in anhydrous DMF (~100 mL), and COS (1.40 g) in DMSO was added dropwise to the reaction mixture. The reaction mixture was added with a few drops of triethylamine and stirred at room temperature overnight. The mixture was purified by dialysis to deionized water for two days. The product was collected and freeze-dried to afford F-COS conjugate as a white powder.

### 4.3. Characterization of Florfenicol-COS Conjugates (F-COS)

#### 4.3.1. Characterization of F-COOH

For ^1^H NMR spectral analysis, samples were dissolved in DMSO-d6 (10 mg/mL). The spectra were recorded using a Bruker-500 NMR spectrometer (Bruker, Bremen, Germany) at 298 K.

#### 4.3.2. Characterization of F-COS

The conjugation product of florfenicol and COS was determined by mass spectrometry. In brief, 1 mg F-COS was dissolved in 1 mL water solution and filtered with 0.22 µM filter. 2 µL sample was then mixed with the same volume of 2, 5-dihydroxybenzoic acid (Sigma-Aldrich, St. Louis, MO, USA) and air-dried for mass spec analysis. The analysis was performed on an autoflex Ш smartbeam MALDI-TOF mass spectrometry in the positive ion mode (Bruker, Bremen, Germany).

### 4.4. Biofilm Inhibition and Eradication Assays

*Streptococcus hyovaginalis* was inoculated in MRS medium for overnight culture and diluted to ~1 × 10^7^ CFU/mL in fresh medium. Biofilms were culture in 96-well polystyrene microtiter plates as previously described [31]. Briefly, 100 μL diluted cell culture was added to each well of sterile flat-bottom 96-well polystyrene microtitre plates and incubated at 30 °C for 24 h. For the biofilm inhibition test, antimicrobial compounds were added together with the cell culture. For the biofilm eradication assay, the microtiter plates were incubated at 30 °C for 24 h to allow biofilm formation. Established biofilms were then incubated at 30 °C in MRS supplemented with compounds as indicated. Cell viability in the biofilm was evaluated by MTT assay. All tests were performed in 6 replicates for each treatment, and all assays were performed three times.

### 4.5. Planktonic Bacteria Inhibition Assay

A volume of 4 mL of *Streptococcus hyovaginalis* (10^5^ CFU/mL) in the MRS medium was incubated with serial dilutions of the antimicrobial compound as indicated at 30 °C with shaking for 24 h. Afterward, cell suspensions were spread onto MRS plates and incubated at 30 °C for 24 h. Colony-forming units (CFU) were counted for each plate.

### 4.6. Fluorescence Microscopy

A volume of 3 mL of 10^7^ CFU/mL bacterial solution was cultured in a sterile 35 mm × 10 mm Cell Culture Dish (Corning, NY, USA) at 30 °C for 24 h. The supernatant was then removed and washed with PBS. Existing biofilms were treated with compounds as above. Samples were then stained with 4,6-diamidino-2-phenylindole (DAPI), and fixed with 4% paraformaldehyde. The fluorescent microscopic images were taken by CLSM TCS SP5 (Leica, Weztlar, Germany).

### 4.7. Statistical Analysis

Graphical evaluations were performed with GraphPad Prism v5.01 (GraphPad Software Inc., San Diego, CA, USA). Analysis of variance (ANOVA) was used to evaluate significant differences. Data are presented as means ± SD. A two-tailed Student’s t-test was performed to compare two groups and one-way ANOVA for multiple group analysis. The *p*-value < 0.05 or 0.01 was considered to be significant. All data were analyzed using IBM SPSS Statistics 19.0 (SPSS Inc., Chicago, IL, USA).

## Figures and Tables

**Figure 1 molecules-25-06043-f001:**
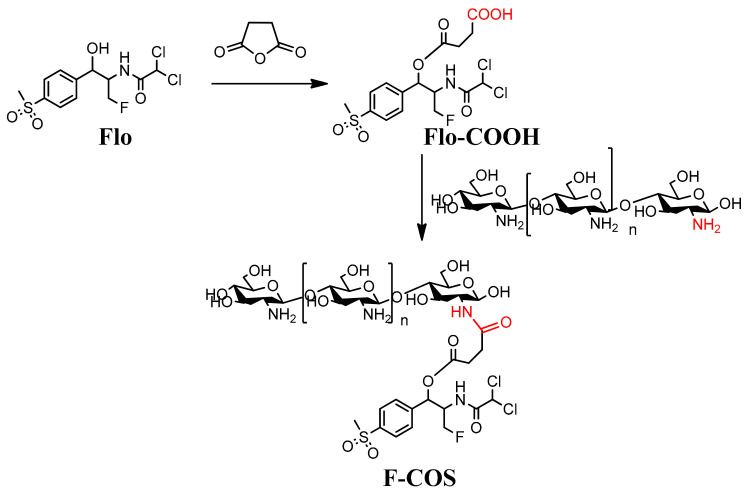
Synthetic routes of Florfenicol-COS conjugates (F-COS). The red color represented the carboxylic acid group of FCOOH condensated with the amino group of COS to form the product F-COS.

**Figure 2 molecules-25-06043-f002:**
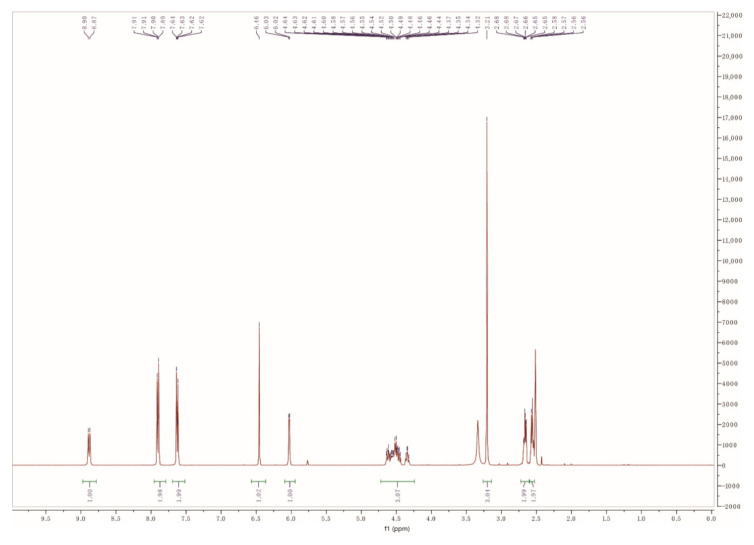
^1^H Nuclear Magnetic Resonance (NMR) characterization of Flo-COOH, the spectra were recorded at 298 K on a Bruker-500 NMR spectrometer. DMSO-d6 (2.5 ppm and 3.3 ppm) was used as the chemical-shift reference in NMR experiments.

**Figure 3 molecules-25-06043-f003:**
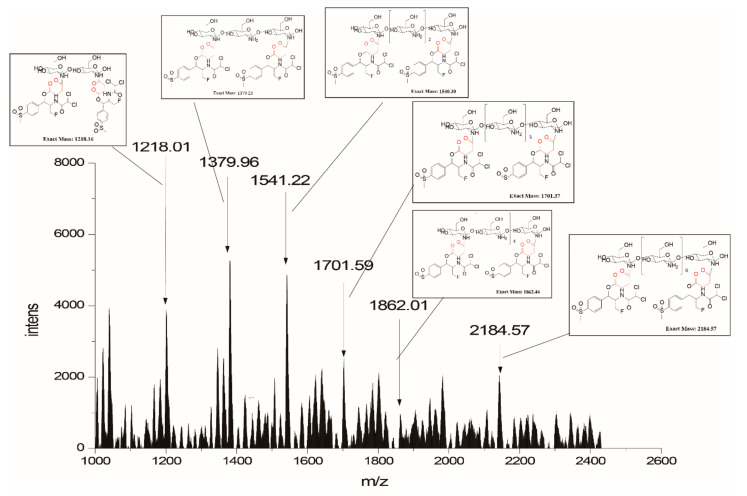
The characterization of F-COS conjugates by MALDI-TOF mass spectrometry in the positive ion mode. 2, 5-dihydroxybenzoic acid was used as the sample matrix.

**Figure 4 molecules-25-06043-f004:**
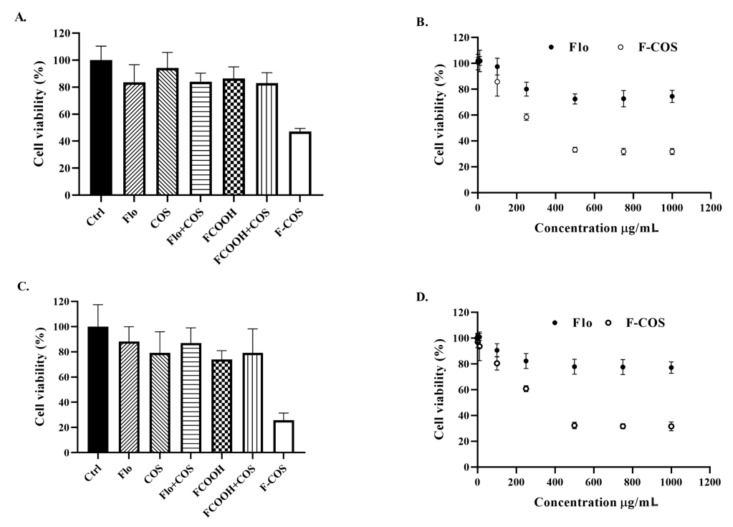
F-COS eradicated *Streptococcus hyovaginalis* biofilms. The anti-biofilm activity of F-COS, florfenicol (Flo), COS, a 1:1 mixture of florfenicol and COS (Flo + COS), Flo-COOH (FCOOH), a 1:1 mixture of FCOOH and COS (FCOOH + COS) were tested against mature biofilm of *Streptococcus hyovaginalis* (**A**) or planktonic cells cultured in biofilm formation condition (**C**), at a concentration of 250 μg/mL. The concentration-dependent of anti-biofilm activity of F-COS against mature biofilms (**B**) or inhibiting biofilm formation (**D**) were shown. Data are represented as means ± SD (n = 6) of cell viability compared to the untreated group.

**Figure 5 molecules-25-06043-f005:**
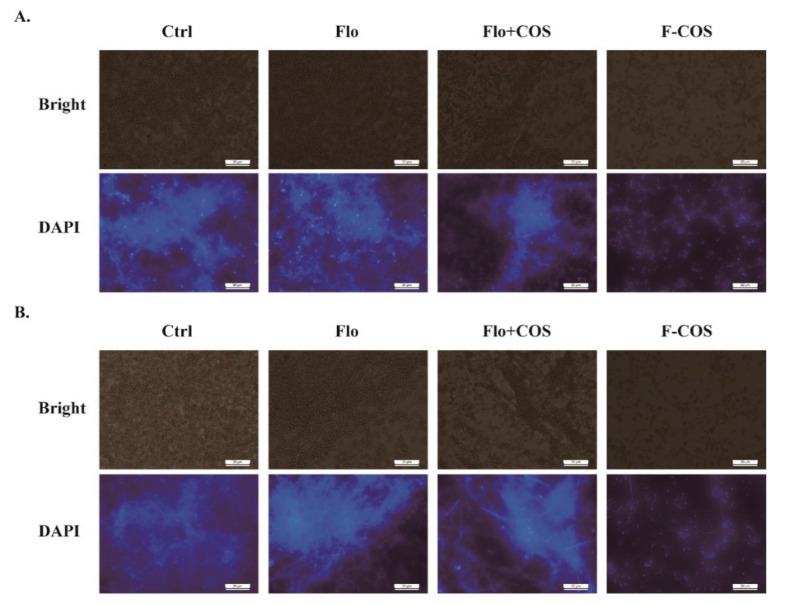
Micrographs of *Streptococcus hyovaginalis* biofilm cells treated with F-COS. Mature Biofilm (**A**) or planktonic cells cultured in biofilm formation condition (**B**) exposure to 0.25 mg/mL F-COS, florfenicol (Flo), a 1:1 mixture of florfenicol and COS (Flo + COS) were stained with 1 µg/mL DAPI and observed under confocal microscopy. Scale bars, 20 μm.

**Figure 6 molecules-25-06043-f006:**
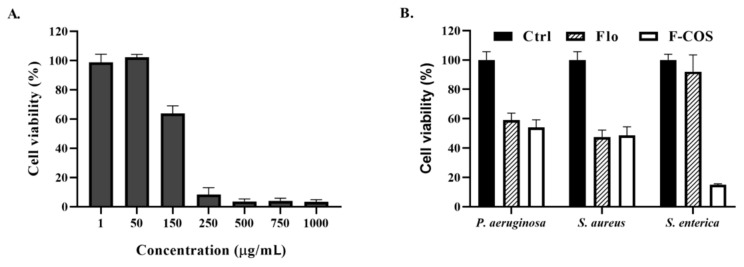
F-COS showed Broad-spectrum antibacterial activities. The antibacterial activity of serial dilutions of F-COS against vegetative-growth *Streptococcus hyovaginalis* cells (**A**). The anti-biofilm activity of F-COS and florfenicol (Flo) was tested against the mature biofilm of *Pseudomonas aeruginosa, Staphylococcus aureus, and Salmonella enterica* at a concentration of 250 μg/mL (**B**). Data are represented as means ± SD (*n* = 6) of cell viability compared to the untreated group.

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
