# Peer review of "Chitosan Oligosaccharides Coupling Inhibits Bacterial Biofilm-Related Antibiotic Resistance against Florfenicol"

_molecules, 2020, doi:10.3390/molecules25246043_

Round 1

Reviewer 1 Report

This work focus on the assumption that chitosan oligosaccharides coupling inhibits bacterial biofilm-related antibiotic resistance against florfenicol. Overall, the work is well designed and written but the discussion should give a comparative view with previous works to clearly demonstrate the advantage of this approach.

More importantly, the authors propose a therapeutic strategy but they cannot clearly explain the mechanism behind it as stated in line 154: "The possible mechanism for the observed results remains unclear. One possible explanation is that the modification on florfenicol with COS impaired the drug-resistant machinery." This explanation is arguable, since the antimicrobial properties of chitosan are widely described, so the possible explanation is the action of chitosan. This is a core issue of the paper, since it is not feasible to propose a therapeutic strategy without understanding the mechanism behind it.

Minor issues to address are that the abstract should highlight the most important quantitative results of the paper, and some references are wrongly sticked together with the text.

Author Response

We greatly appreciate the thoughtful efforts of the reviewers. As detailed below, we have carefully revised the manuscript and figures to address the reviewer suggestions.

  1. More importantly, the authors propose a therapeutic strategy, but they cannot clearly explain the mechanism behind it as stated in line 154: "The possible mechanism for the observed results remains unclear. One possible explanation is that the modification on florfenicol with COS impaired the drug-resistant machinery." This explanation is arguable, since the antimicrobial properties of chitosan are widely described, so the possible explanation is the action of chitosan. This is a core issue of the paper, since it is not feasible to propose a therapeutic strategy without understanding the mechanism behind it.

We appreciated this advice and have now expanded our discussions of the mechanism regarding the properties of chitosan in the Discussion (Line 171-177).

  1. Minor issues to address are that the abstract should highlight the most important quantitative results of the paper, and some references are wrongly sticked together with the text.

We acknowledge theses points and have accordingly revised the Abstract (Line 19-22) and corrected the references.

Reviewer 2 Report

Authors of the manuscript entitled “Chitosan oligosaccharides coupling inhibits bacterial biofilm-related antibiotic resistance against florfenicol” demonstrate the concept that by conjugation of the antibiotic to a defined oligosaccharide it can be delivered in a more efficient way, lowering the doses required and thus achieving more efficient eradication of the biofilm forming bacteria. The presented reasoning suggests the potential of such conjugates in treatments of the bio-film-related infections. The manuscript is properly structured, the methods and techniques used are up-to-date and relevant for this type of studies. However, it is not free from errors and some ambiguities that I point out in my specific comments.

Some specific comments:

1) What was the solvent used in the NMR measurements – the DMSO-d6 is mentioned in the text whereas in the legend to the Figure 2 there are both DMSO-d6 and D2O? Similarly, check the Materials and Methods section.

2) Figure 2 – the integrals should be moved to the level of peaks they actually indicate/represent (numerical peak indicators could be removed). Were there any features characteristic for the Flo-COOH in the spectrum [change upon introduction of the COOH group?].

3) The mass of the Flo-COOH should be mentioned in the text, as well as the calculated number of the GlcN residues in the oligosaccharides as indicated in the MALDI-TOF spectrum [Figure 3]. What was the reference spectrum of COS prior to conjugation like? Mode (positive or negative?) and the matrix used should be indicated in the legend to this figure and in the Materials and Methods section.

4)Line 86 – the brief description of the actual assay should be added.

5) Figure 4 and Figure 6 – in all the panels of the figure the cell viability is expressed as “%”, but the values on the Y-axes do not seem to be the percentages? Please verify and correct.

6) As demonstrated by the Authors each oligosaccharide unit was substituted by two Flo-residues? Was this substitution at random positions in the oligosaccharides larger than a disaccharide? Or were Flo-residues always at both reducing and non-reducing ends? [like in the COS disaccharide].

7)There was a 4-fold improvement of the F-COS vs. Flo alone in the observed inhibition of the biofilm formation. It seems that the effective drug concentration at the bacterial interface has changed. Could it be due to an increased solubility of the florfenicol in the conjugated form? This aspect could be added to discussion.

8) Line 221 – “[...] The supernatant was removed and washed with PBS.” I guess the plate with the biofim on it was washed with PBS, and not the supernatant?

9) The text should be double-checked for typing errors, missing subscripts, and non-italicized names of bacteria; Line 153 – a lost sentence, or?

Author Response

We greatly appreciate the thoughtful efforts of the reviewers. As detailed below, we have carefully revised the manuscript and figures to address the reviewer suggestions.

  1. What was the solvent used in the NMR measurements – the DMSO-d6 is mentioned in the text whereas in the legend to the Figure 2 there are both DMSO-d6 and D2O? Similarly, check the Materials and Methods section.

The solvent used in the NMR is the DMSO-d6. We are really sorry for our negligence. We have revised the text in the Results (Line 75-76) and Materials and Methods (Line 217-218).

  1. Figure 2 – the integrals should be moved to the level of peaks they actually indicate/represent (numerical peak indicators could be removed). Were there any features characteristic for the Flo-COOH in the spectrum [change upon introduction of the COOH group?].

We appreciated the suggestions. We have added the text regarding the features characteristic for the Flo-COOH in the spectrum in the Results (Line 71-73). The peaks from 2.46 to 2.56 in the NMR spectrum indicated the methylene groups of butanedioic acid, suggesting the successful conjugation of florfenicol and butanedioic acid.

  1. The mass of the Flo-COOH should be mentioned in the text, as well as the calculated number of the GlcN residues in the oligosaccharides as indicated in the MALDI-TOF spectrum [Figure 3]. What was the reference spectrum of COS prior to conjugation like? Mode (positive or negative?) and the matrix used should be indicated in the legend to this figure and in the Materials and Methods section.

In response to these suggestions we now added the mass of the Flo-COOH and calculated number of the GlcN residues in COS in the text (Line 81-83). We also include a mass spectrum of unmodified COS in the supplementary data. The Mass mode we used here is positive ion mode, and the matrix used was 2,5-dihydroxybenzoic acid. Accordingly, we have revised the legends to Figure 3 (Line 89-90) and the related text in the Materials and Methods section (Line 224).

  1. Line 86 – the brief description of the actual assay should be added.

As suggested, we have added the description of the assay in the result (Line 93-95).

  1. Figure 4 and Figure 6 – in all the panels of the figure the cell viability is expressed as “%”, but the values on the Y-axes do not seem to be the percentages? Please verify and correct.

As suggested, we have revised Figure 4 and Figure 6.

  1. As demonstrated by the Authors each oligosaccharide unit was substituted by two Flo-residues? Was this substitution at random positions in the oligosaccharides larger than a disaccharide? Or were Flo-residues always at both reducing and non-reducing ends? [like in the COS disaccharide].

Yes, we found that each oligosaccharide unit was substituted by two Flo-residues in our experimental setting. We have done some preliminary explorations on the substitution pattern of Flo on the chitosan oligosaccharide. It seems that Flo residues were not only at reducing and non-reducing end, but happens at random positions in the larger oligosaccharide. However, our preliminary MS/MS result of Flo-COS was too complicated to interpret. We’d like to use highly purified COS penta-saccharide and tetra-saccharide to conjugate with Flo to address this question in our future study.

  1. There was a 4-fold improvement of the F-COS vs. Flo alone in the observed inhibition of the biofilm formation. It seems that the effective drug concentration at the bacterial interface has changed. Could it be due to an increased solubility of the florfenicol in the conjugated form? This aspect could be added to discussion.

We appreciated this advice and now address this possibility in the Discussion (Line 177-179).

  1. Line 221 – “[...] The supernatant was removed and washed with PBS.” I guess the plate with the biofilm on it was washed with PBS, and not the supernatant?

The biofilm experiment was performed according to the method introduced in reference 32. “Pitts, B., et al., A microtiter-plate screening method for biofilm disinfection and removal. Journal of Microbiological Methods, 2003. 54(2): p. 269-276.”. We revised th text (Line 227-228) and added this reference in the manuscript.

  1. The text should be double-checked for typing errors, missing subscripts, and non-italicized names of bacteria; Line 153 – a lost sentence, or?

We are really sorry for our negligence. We have double-checked the typing errors, adding subscripts and italicized name for bacteria; The sentence of Line 153 in the original document has been completed (Line 164-165).